## [Transparent Peer Review file · Nature Communications]

A heterogeneous population code at the first synapse of vision

Corresponding Author: Professor Tom Baden

Version 0:

Reviewer comments:

Reviewer #1

(Remarks to the Author)

This is an interesting and exciting study that increases our understanding of early visual processing by the retina. They show heterogeneity of synaptic glutamate release at cone synapses that has important functional implications for early processing of vision. To do this, they expressed a glutamate sensor, iGluSnFr, in zebrafish horizontal cells and selectively stimulated red-sensitive cones with an amber light stimulus. By performing this with intact fish, they avoid variability that can arise from differences in the health of individual cells when one studies isolated retina. They build on their experimental results by modeling the effects of synaptic changes in contrast sensitivity on visual behavior. They used a range of sophisticated stimuli and analytical techniques to analyze their data. It is of the highest technical quality. To put the work in better context, I suggest a number of key relevant citations. I also have questions about the potential impact of glutamate buffering by iGluSnFr and measurement noise on their results, along with a few smaller questions.

1. A major question of mine is how much of the measurement variability is due to technical measurement variability of small fluorescent signals and how much is genuine biological variability. Please discuss how the authors assessed technical variability to address this issue.
2. The authors should consider and discuss the potential impact of glutamate buffering by iGluSnFr in horizontal cells. In this context, they can cite Armbruster, Dulla, and Diamond, 2020, eLife.
3. Line 257. In addition to measurement noise, to what extent are the smaller responses to light increments due to the limited ability of iGluSnFr to report decrements in release? The authors discuss the smaller signal to noise for responses to light increments on line 155 when explaining why they studied off steps. This limited signal to noise for light flashes is also relevant for later measurements in which they suggest greater sensitivity to light off than light on.
4. To put these studies in a proper context of prior work, I suggest they note that Burkhardt and colleagues previously showed a tremendous (up to 20fold) increase in contrast sensitivity across the cone synapse by comparing cone vs. bipolar cell contrast responses (Burkhardt and Fahey, J. Neurophysiol. 1998). They also showed that the range of contrast sensitivities measured in bipolar cells closely matched the range of contrasts present in real world images (Burkhardt, Fahey and Sikora, Vis Neurosci, 2006). Furthermore, this increase in contrast sensitivity is present in synaptic currents of both ON and OFF bipolar cells suggesting it arises presynaptically (Thoreson and Burkhardt, Vis Neurosci 2003), consistent with present results.
5. When discussing the role of horizontal cell feedback in modulating contrast sensitivity (line 678), I also suggest they cite work by Fahey and Burkhardt who showed evidence that the increase in contrast sensitivity at cone synapses can be modulated by horizontal cell feedback (Fahey and Burkhardt, Vis Neurosci 2001), consistent with the present results.
6. Page 5. In contrast to the wide variations in amplitude/intensity relationships seen in Fig. 2, amplitude/intensity relationships for cone light responses are essentially identical among cones with the same spectral sensitivity (e.g., Baylor and Fuortes, J Physiol 1970). I suggest the authors note this fact and contrast it with heterogeneity in sensitivity measurements of Fig. 1 to highlight further transformations at the cone synapse.
7. I suggest including additional discussion of mechanisms that might contribute to cone-to-cone variations in synaptic

release. In that context, Grassmeyer and Thoreson (Front Cell Neurosci 2019) combined electrophysiology and calcium imaging at individual cone ribbons to show that while the voltage-dependence of whole cell calcium currents is quite similar among cones, there is greater variability in the voltage-dependence of calcium signals measured at individual ribbons. This study directly showing ribbon-to-ribbon variability could be cited when discussing potential sources of heterogeneity (e.g., line 417). Another potential contributor to cone-to-cone variability in synaptic output that can also be mentioned is variation among cones in their resting membrane potentials (and this has never been carefully measured).

8. Using paired recordings and confocal calcium imaging, Grassmeyer and Thoreson also showed directly that a single horizontal cell can modify calcium signals at one ribbon independent of another in the same cone. This study can be cited as additional strong evidence for the role of horizontal cells in directly adjusting synaptic sensitivity (e.g., line 678).

9. Did blocking feedback alter variability in D1/2 measurements? If the greater heterogeneity in D1/2 values of Fig. 2 arises at the synapse, then one would predict that it would also be sensitive to blockade of feedback. If not, then I would be more concerned that it might reflect measurement variability rather than biological variability.

10. Line 179. I suggest explicitly stating that the range of flash durations used in Fig. 2 fall within the integration period of an individual cone where flash duration and intensity are linearly related to one another (Baylor and Hodgkin, 1973; J Physiol). In this temporal range, the half maximal duration can thus be related directly to half maximal intensity.

11. Line 264, 287, 636. I suggest citing the highly relevant study by Jackman et al. (Nature Neurosci 2009) who showed that the prominent off responses seen in horizontal cells following an OFF step arise from the replenishment of vesicles to the releasable pool on the ribbon during the prior period of illumination (and thus hyperpolarization of the cone).

12. While it is likely that the major effects seen in the present study are due to blocking negative feedback, I suggest that the authors note that CNQX will also block positive feedback from horizontal cells (Jackman et al., PLoS Biol 2011). If they wish to disentangle these things in future studies, the authors may wish to use HEPES to selectively block negative feedback.

13. Why average only 12 trials for the homogenous model but 1000 for the heterogeneous? While I can see there might be less of a need for a large sample when using a homogenous population, that is an unexpectedly huge difference. Is any of the improved performance of the heterogeneous vs. homogeneous model in Fig. 7F simply due to averaging 1000 trials for the former but only 12 for the latter?

Reviewer #2

(Remarks to the Author)

This is a beautiful study by Herzog et al which addresses fundamental questions of signal encoding at the photoreceptor synapse in vivo. The authors use in vivo two-photon optical imaging of glutamate release from putative single cone photoreceptor terminals in zebrafish larval retina and identify unique features of cone synapse function. The authors identify several interesting aspects of cone synapse function - i) high reliability and precision of signal encoding that deviates from the classical stochastic (Poisson) process of neurotransmitter release in conventional synapses; ii) heterogeneity in luminance, contrast and temporal encoding between synapses of neighboring cones with more pronounced encoding of light decrements than increments and iii) local heterogeneity in output of cone synapse is caused due to HC feedback. In addition, the authors develop a model of cone convergence on bipolar cells showing that a heterogenous population of cone signals is better suited for encoding variations in contrast encountered during natural vision than a homogenous population. Overall, the results are exciting, the data is solid, figures and text well presented. However, there are a few remaining issues outlined below which need to be addressed to bolster the significance of this story.

1. How does the reliability and temporal precision of cone output vary with the mean release rate? Can the authors use a slightly lower background light level and test if both reliability and temporal precision is poorer with higher baseline vesicle release as one might expect?

2. Why does cone output response to lower flash durations increase the temporal jitter by ~3-fold compared to 40 ms flash durations? Is this simply a signal to noise issue and perhaps an ambiguity in detecting the peak of the cone response for a weaker stimulus? Does the cone response to shorter flash durations also exhibit a low variance across trials?

3. From the cone responses to 20Hz flickering stimuli, it seems that some cone output synapses can reliably track signals and may exhibit a faster rate of recovery from synaptic depression. Can this timescale of synaptic recovery be measured by paired light flashes of varying intervals?

4. Does the strength of temporal tuning correlate with the DLI across cone output synapses?

5. HC feedback should also affect the kinetics of cone output responses to light decrement and increment flashes. This should be tested in addition to any potential effect of HC feedback on luminance coding, reliability and temporal precision of cone output.

6. The DLI of cone output should be highly dependent on the background light level. More hyperpolarized cones with lower baseline release rate should have a more negative DLI. This should be directly shown. This also suggests that the effect of

HC feedback on DLI will be less at lower or moderate background light level compared to brighter backgrounds.

Version 1:

Reviewer comments:

Reviewer #1

(Remarks to the Author)

My earlier questions and concerns have all been adequately addressed in this interesting and technically impressive study.

Reviewer #2

(Remarks to the Author)

The authors have done a great job addressing all of my concerns.

I have a minor comment about mentioning and citing some relevant literature about the asymmetry in cone responses to negative versus positive contrasts. In line 607, the authors can add citations to recent work (Angueyra et al J Neurosci 2021 and Saha et al Nature Comm 2024) in primate cones which show that contrast asymmetry within individual cones is already quite prominent at the level of transduction, as seen in the photocurrent and photovoltage responses which show greater sensitivity to negative than positive contrasts, but as nicely shown in this study the cell-to-cell variation in contrast asymmetry across cone population may be created at the synaptic level.

Overall the revisions to the manuscript have significantly strengthened the story and provided new mechanistic insights. I have no further comments.

General preamble to all reviewers

We thank both reviewers for critically engaging with our work and for their valuable suggestions. In response, we have performed new experiments and analysis and rewritten several parts of the main text. Additions include:

1. New experiments explore heterogeneity in the kinetics of vesicle refilling using a paired-pulse protocol, shown in a new Figure 6.
2. New experiments test the effect of different levels of luminance on release from cones, shown in the new Supplemental Figure S4.
3. New experiments deal with minor points, including biosensor expression levels (which are quite homogeneous) and heterogeneity in DLI across luminance levels.
4. New analyses of various datasets which confirm and extend previous assertions.
5. Editing for clarity and addition of various references, in line with reviewer suggestions

In addition, we have updated the nomenclature to describe ‘red’ cones to align with the new naming system put forward in a recent consensus paper, aimed at aligning cone-naming systems across vertebrates (Baden et al. 2025 PLoS Biol). In the new system, zebrafish ‘ancestral red single cones’ correspond to ‘type 1 photoreceptors’, or PR1. The new naming system highlights the deep homology of vertebrate cones and illustrates that zebrafish PR1 is homologous to mouse and human PR1 (previously “green/M” cones in mouse, and red/green; L/M-cones in human):

Figure 1 from PLoS Biology 2025 consensus view; A standardized nomenclature for the rods and cones of the vertebrate retina. Baden et al.

We hope the reviewers will find these revisions useful and suitable to address all points raised, and we look forward to hearing back in due course.

Reviewer #1

This is an interesting and exciting study that increases our understanding of early visual processing by the retina. They show heterogeneity of synaptic glutamate release at cone synapses that has important functional implications for early processing of vision. To do this, they expressed a glutamate sensor, iGluSnFr, in zebrafish horizontal cells and selectively stimulated red-sensitive cones with an amber light stimulus. By performing this with intact fish, they avoid variability that can arise from differences in the health of individual cells when one studies isolated retina. They build on their experimental results by modeling the effects of synaptic changes in contrast sensitivity on visual behavior. They used a range of sophisticated stimuli and analytical techniques to analyze their data. It is of the highest technical quality.

We thank the reviewer for their encouraging engagement with our work and for their valuable suggestions.

To put the work in better context, I suggest a number of key relevant citations. I also have questions about the potential impact of glutamate buffering by iGluSnFr and measurement noise on their results, along with a few smaller questions.

Thank you – we have aimed to respond to all points raised in full.

1. A major question of mine is how much of the measurement variability is due to technical measurement variability of small fluorescent signals and how much is genuine biological variability. Please discuss how the authors assessed technical variability to address this issue.

We thank the reviewer for raising this important point. In short, we are confident that our reported functional heterogeneity across the population of cones relates to meaningful biological variation rather than measurement noise. Even though heterogeneity is evident in single trials within individual cones, the metrics of variability across the population are all based on averaged measurements across many cones. In each cone, many trials were used to obtain that average (typically between 10 and 98). Properties measured in this way include contrast-response functions (Fig 2), amplitudes, CV and temporal jitter (Fig 3), DLI (Fig 4) and baseline (Fig. 6). Additionally, several of these metrics are normalised to an individual cone's response range before comparison.

The variability between cones was not due to variations in expression of the biosensor. We performed new experiments staining against GFP in Tg(cx55.5:SFiGluSnFR) and these showed much more consistent levels of expression across the HC dendritic terminals invaginating cone terminals compared to SFiGluSnFR signal observed under two-photon in live tissue (Fig. 1 below).

Reviewer figure 1 | *Tg(cx55.5:SFiGluSnFR)* expression is variable in vivo under two-photon conditions, but not in fixed tissue under confocal. **A**, A typical experimental view of *Tg(SFiGluSnFR)* in HCs of a 6 dpf larval zebrafish in the nasal retina of a living larval zebrafish, in a sagittal plane. HC dendritic connections with pre-synaptic cones are recognised by their globular structure in the distal OPL, and are circled (red, dashed). As shown in the data throughout the paper, the fluorescence signals at HC tips, reflecting pre-synaptic cone activity, are heterogeneous between nasal PR1, as well as between cone types. **B**, Fixed, flat-mount retina expressing *Tg(cx55.5:SFiGluSnFR)* of a 6 dpf larval zebrafish, immunolabeled against GFP. HC dendritic projections to pre-synaptic cones are recognisable by their rosette patterning, and are circled (red, dashed). Note the heterogeneity in reporter expression between cone terminals is much reduced compared to the same transgenic line in vivo, under two-photon.

2. The authors should consider and discuss the potential impact of glutamate buffering by iGluSnFr in horizontal cells. In this context, they can cite Armbruster, Dulla, and Diamond, 2020, eLife.

We agree that as with any exogenously expressed proteins, overexpression might disrupt cellular function (e.g. here, by competing with endogenous glutamate signals). However, we do not think that putative overexpression can explain our results. As noted above (point 1), baseline fluorescence across PR1 in vivo was heterogeneous while immunolabelling for the GFP within the SFiGluSnFR showed no such heterogeneity. Accordingly, it appears that horizontal cells including their tips express similar levels of SFiGluSnFR, at a level that permits a substantial range of physiological heterogeneity.

We now note this in lines 778ff, alongside the suggested citation.

3. Line 257. In addition to measurement noise, to what extent are the smaller responses to light increments due to the limited ability of iGluSnFr to report decrements in release? The authors discuss the smaller signal to noise for responses to light increments on line 155 when explaining why they studied off steps. This limited signal to noise for light flashes is also relevant for later measurements in which they suggest greater sensitivity to light off than light on.

The reviewer is referring to the analysis in Figure 4 (Differential encoding of positive and negative contrasts). The smaller responses to light increments are not affected by the lower SNR for decreasing iGluSNFR signals because they are not single trials – they are averages of multiple trials. The example traces in Fig. 4C and D are averages

of 10 trials within individual cones and the collected results in Fig. 4E and F are from 90 cones. The smaller responses to light increments that we report are therefore based on measurements made after averaging out noise.

4. To put these studies in a proper context of prior work, I suggest they note that Burkhardt and colleagues previously showed a tremendous (up to 20fold) increase in contrast sensitivity across the cone synapse by comparing cone vs. bipolar cell contrast responses (Burkhardt and Fahey, J. Neurophysiol. 1998). They also showed that the range of contrast sensitivities measured in bipolar cells closely matched the range of contrasts present in real world images (Burkhardt, Fahey and Sikora, Vis Neurosci, 2006). Furthermore, this increase in contrast sensitivity is present in synaptic currents of both ON and OFF bipolar cells suggesting it arises presynaptically (Thoreson and Burkhardt, Vis Neurosci 2003), consistent with present results.

We have extensively investigated the transmission of the visual signal through bipolar cells in the retina of larval zebrafish. We agree that there is, on average, an increase in contrast-sensitivity of bipolar cells compared to cones and now note this with the first Burkhardt reference as well as one of our own which is specific to zebrafish (text relating to Fig. 4F). However, the increase in contrast-sensitivity does not arise pre-synaptically, at least in zebrafish. In unpublished experiments, we have used iGluSnFR to simultaneously measure the synaptic inputs and outputs of individual bipolar cells and find that the saturation of the output at higher contrasts is caused by inhibition through the amacrine cell network in the IPL.

5. When discussing the role of horizontal cell feedback in modulating contrast sensitivity (line 678), I also suggest they cite work by Fahey and Burkhardt who showed evidence that the increase in contrast sensitivity at cone synapses can be modulated by horizontal cell feedback (Fahey and Burkhardt, Vis Neurosci 2001), consistent with the present results.

Thank you, we could not find the 2001 Vis Neuro paper as noted but there is the 2003 one by the same authors/journal ("Center-surround organisation in bipolar cells..") which partly deals with horizontal cell feedback. We suspect this is the one the reviewer was referring to? Added as suggested.

6. Page 5. In contrast to the wide variations in amplitude/intensity relationships seen in Fig. 2, amplitude/intensity relationships for cone light responses are essentially identical among cones with the same spectral sensitivity (e.g., Baylor and Fuortes, J Physiol 1970). I suggest the authors note this fact and contrast it with heterogeneity in sensitivity measurements of Fig. 1 to highlight further transformations at the cone synapse.

Done.

7. I suggest including additional discussion of mechanisms that might contribute to cone-to-cone variations in synaptic release. In that context, Grassmeyer and Thoreson (Front Cell Neurosci 2019) combined electrophysiology and calcium imaging at individual cone ribbons to show that while the voltage-dependence of whole cell calcium currents is quite similar among cones, there is greater variability in the voltage-dependence of calcium signals measured at individual ribbons. This study directly showing ribbon-to-ribbon variability could be cited when discussing potential sources of heterogeneity (e.g., line 417). Another potential contributor to cone-to-cone variability in synaptic output that can also be mentioned is variation

among cones in their resting membrane potentials (and this has never been carefully measured).

Done.

8. Using paired recordings and confocal calcium imaging, Grassmeyer and Thoreson also showed directly that a single horizontal cell can modify calcium signals at one ribbon independent of another in the same cone. This study can be cited as additional strong evidence for the role of horizontal cells in directly adjusting synaptic sensitivity (e.g., line 678).

Added as suggested.

9. Did blocking feedback alter variability in D1/2 measurements? If the greater heterogeneity in D1/2 values of Fig. 2 arises at the synapse, then one would predict that it would also be sensitive to blockade of feedback. If not, then I would be more concerned that it might reflect measurement variability rather than biological variability.

We did not record responses to the mixed off-step duration stimulus before and after HC block, therefore we cannot comment how stimulus-response functions may shift after HC blockade, and how this would affect heterogeneity at the population level. However, we do show how contrast-response functions change after HC block, and that at the population level the extracted response metrics of DLI and baseline release tend to decrease in variability (Fig 6, B, D and E).

10. Line 179. I suggest explicitly stating that the range of flash durations used in Fig. 2 fall within the integration period of an individual cone where flash duration and intensity are linearly related to one another (Baylor and Hodgkin, 1973; J Physiol). In this temporal range, the half maximal duration can thus be related directly to half maximal intensity.

Done.

11. Line 264, 287, 636. I suggest citing the highly relevant study by Jackman et al. (Nature Neurosci 2009) who showed that the prominent off responses seen in horizontal cells following an OFF step arise from the replenishment of vesicles to the releasable pool on the ribbon during the prior period of illumination (and thus hyperpolarization of the cone).

Done.

12. While it is likely that the major effects seen in the present study are due to blocking negative feedback, I suggest that the authors note that CNQX will also block positive feedback from horizontal cells (Jackman et al., PLoS Biol 2011). If they wish to disentangle these things in future studies, the authors may wish to use HEPES to selectively block negative feedback.

Thanks, to explore this we did use HEPES as an alternative to CNQX, with essentially the same results – see Supplemental Figure S3. This additional dataset was however insufficiently flagged in the previous MS (now been addressed, line 441/2).

13. Why average only 12 trials for the homogenous model but 1000 for the heterogeneous? While I can see there might be less of a need for a large sample when using a homogenous population, that is an unexpectedly huge difference. Is any of the improved performance of

the heterogeneous vs. homogeneous model in Fig. 7F simply due to averaging 1000 trials for the former but only 12 for the latter?

Thanks for raising this. The 1,000 iterations were chosen to reasonably capture the cone-combinatorial space to map the 12 recorded cones onto all possible combinations of five. By contrast, the combinatorial space to map the 12 cones homogeneously is 12 by definition).

The full heterogeneous combinatorial space equals 792:

$$C(n,k) = n! / (k!(n-k)!),$$

Where $n = 12$ (total datasets) and $k = 5$ (number chosen at a time)

$$\text{So: } 12! / (5!(12-5)!) = 12! / (5!*7!) = 95040/120 = 792.$$

Changes to this iteration number did not qualitatively affect the results.

Reviewer #2

This is a beautiful study by Herzog et al which addresses fundamental questions of signal encoding at the photoreceptor synapse in vivo. The authors use in vivo two-photon optical imaging of glutamate release from putative single cone photoreceptor terminals in zebrafish larval retina and identify unique features of cone synapse function. The authors identify several interesting aspects of cone synapse function - i) high reliability and precision of signal encoding that deviates from the classical stochastic (Poisson) process of neurotransmitter release in conventional synapses; ii) heterogeneity in luminance, contrast and temporal encoding between synapses of neighboring cones with more pronounced encoding of light decrements than increments and iii) local heterogeneity in output of cone synapse is caused due to HC feedback. In addition, the authors develop a model of cone convergence on bipolar cells showing that a heterogenous population of cone signals is better suited for encoding variations in contrast encountered during natural vision than a homogenous population. Overall, the results are exciting, the data is solid, figures and text well presented.

We thank the reviewer for critically engaging with our work and their support.

However, there are a few remaining issues outlined below which need to be addressed to bolster the significance of this story.

1. How does the reliability and temporal precision of cone output vary with the mean release rate? Can the authors use a slightly lower background light level and test if both reliability and temporal precision is poorer with higher baseline vesicle release as one might expect?

Thank you for raising this interesting point, which we now address in new experiments showing how amplitude, reliability and temporal precision remain high across all different light conditions. This new data is now included and discussed as Supplemental Figure S4.

Specifically, we imaged cone glutamate responses to trains of off-steps at three different background light intensities. The recordings were continuous, and an electronic filter wheel containing neutral density filters positioned in the stimulus light path generated different background light intensities for each repeat of the off-step train (A). ND 1 (10-fold decrease), ND 0.5 (3.2-fold decrease) and ND 0 (no filter) conditions consistently evoked different sustained glutamate release rates, visibly decreasing with increasing background light level (A, B and D), with correspondingly increased amplitudes (C and E). Across light conditions, cones continued to exhibit highly reliable responses, with variation in response amplitudes consistently falling below Poisson (F,G).

Likewise, the temporal precision of PR1 responses to trains of off-steps of light remains high across all three conditions (H-J), with the majority of cone responses fluctuating in the order of 1-2 ms (J).

Throughout, functional heterogeneity remained a prominent feature of responses across conditions. The dimmest light condition tended to produce responses with the highest amplitude variability, but increasingly brighter background light did not always result in more consistent responses; some cones exhibit lowest CV at ND 0.5, and

others at 1.0. Likewise for temporal precision, ND 0.5 tended to produce responses with the lowest temporal jitter, which may increase in both the dimmer and brighter conditions.

Supplemental Figure S4 | Response amplitude and timing vary with mean light level. **A**, Trains of 40 ms flashes to darkness (30 repeats, 460 ms inter stimulus interval) were delivered at 3 background light levels; neutral density filters (ND) produced a ~10-fold (ND 1.0, black) and ~3.2-fold decrease (ND 0.5, red) from full LED brightness (no ND filter, ND 0, blue). Deconvolved glutamate responses of an example PR1 are shown. **B**, Different light levels evoke glutamate responses with different baselines (dashed lines) and amplitudes (based on A). **C**, Response amplitude measured from baseline glutamate release for each background light condition (mean release during 4 s of steady background light prior to off-step train) to the response peak. Distributions of response amplitudes for each light condition are plotted as histograms (based on A), with Gaussian fits superimposed (ND 1.0, black, mean = 0.8 ± 0.2 ; ND 0.5, red, mean = 1.9 ± 0.3 ; ND 0, blue, mean = 2.4 ± 0.2). **D**, Across 139 PR1 (6 fish), baseline glutamate release decreases with increased background light levels. **E**, Response amplitude tends to increase with background light levels, although heterogeneity in this trend is present between ND 0.5 to

ND 0: 52% of cones exhibited a decrease in response amplitude when background light levels were highest (ND 0). **F**, In all background light conditions, the relationship between mean and standard deviation (SD) of response amplitudes consistently fell below the equivalence line (where mean = SD), as would be expected from a Poisson release process (dashed line). **G**, The coefficient of variance (CV) of the data shown in E and F shows responses at the lowest background light level exhibit significantly higher variance (ND 1.0, black, mean = 0.3 ± 0.1 ; ND 0.5, red, mean = 0.1 ± 0.1 ; ND 0, blue, mean = 0.2 ± 0.1 ; post-hoc Turkey test following a significant one-way ANOVA of $p < .00001$) for pairwise comparisons between ND 1.0 and ND 0.5, and ND 1.0 and ND 0 (both $p < .00001$), but not significant between ND 0.5 and ND 0 ($p = 0.32$). High temporal precision was observed over all three light conditions. Response time was measured as time to threshold (mean glutamate release + 2 SDs), as shown for the example cone responses to the ND 1.0 condition, **H**, and for all PR1 across the three light conditions, **I**, (ND 1.0, black, mean = 30.5 ± 3.1 ; ND 0.5, red, mean = 27.0 ± 1.9 ; ND 0, blue, mean = 28.9 ± 3.6). Temporal precision, quantified as temporal jitter (SD of time to threshold), (ND 1.0, black, mean = 3.4 ± 2.0 ; ND 0.5, red, mean = 1.6 ± 1.0 ; ND 0, blue, mean = 2.4 ± 1.8). Pairwise comparisons of temporal jitter between ND 1.0 and ND 0.5, and ND 0 and ND 0.5 are both significantly different (both $p < .00001$), but not comparison between ND 1.0 and ND 0 ($p = 0.51$) (post-hoc Turkey test following a significant one-way ANOVA of $p < .00001$). As with amplitude, there is heterogeneity in how background light level affects the response time and temporal precision of nasal red-cones, and a general rule is not present. (For D - G, I, J, $n = 139$ red-cones, 6 fish).

2. Why does cone output response to lower flash durations increase the temporal jitter by ~3-fold compared to 40 ms flash durations? Is this simply a signal to noise issue and perhaps an ambiguity in detecting the peak of the cone response for a weaker stimulus? Does the cone response to shorter flash durations also exhibit a low variance across trials?

Thanks for raising this. In response, we reanalysed glutamate release events in response to off-steps of light (both trains of off-steps and mixed off-step duration stimuli – i.e. Figs. 2,3). The results from this are summarised in Reviewer Figure 2 below.

First, changes in glutamate release were only counted as light-evoked responses if they surpassed a threshold, here defined as the mean + 2 SD of the glutamate signal during steady bright light exposure (2 s at start of recording and 2 s at end of recording), when glutamate release is suppressed to almost zero thereby giving a measure of the “noise” in the recording. Using a threshold to count only the responses that rise above the noise threshold in the recording facilitates a more accurate detection of responses, reduces the chance of noise being counted as a response, and reduces ambiguity in response detection for smaller responses.

Second, the timings of responses exceeding threshold were re-analysed using time-to-threshold in place of time-to-half-peak, and temporal jitter (SD) around the threshold in place of temporal jitter around half-peak. The amplitudes of these responses exceeding the threshold were counted as the maximum peak within the response window, as before.

As a result, the amplitude and timing analyses of responses shown in all panels of figure 3 are now updated to reflect a more accurate detection of responses.

To directly answer the reviewers query about responses to shorter off-step durations, we observe that these responses tend to be smaller in amplitude (A) and a larger proportion of responses do not reach the threshold, compared to larger amplitude responses typical of longer off-step duration conditions. The responses that do exceed threshold exhibit a higher variance in amplitude across trials compared to other

conditions, demonstrated by the higher CV value at shorter off-step duration (B and D). However, when plotting the mean versus SD of response amplitudes for all stimulus conditions for all cones, >99% fall below Poisson distribution, indicating that across all conditions measured, the detected response amplitude remains highly reliable (C). Likewise, the temporal precision of responses to shorter off-step is reduced, with a slightly longer time-to-threshold (E) and higher temporal jitter across trials (F).

The results from this more stringent re-analysis of amplitude and timing precision of responses to shorter off-steps align with what is shown in the original figures.

Reviewer figure 2 | Re-analysis of amplitude reliability and temporal precision of responses to mixed-duration off-steps of light. The amplitude (A-D) and temporal precision (E-F) of responses surpassing a set threshold were re-analysed for 70 red-cones, 12 fish. (A) Mean response amplitude (\pm SD) and (B) response coefficient of variance (CV) are plotted as a function of off-step duration. (C) SD of response amplitude over 25 stimulus repetitions is plotted against mean response amplitude for each off-step duration, for each cone. The SD axis is plotted in log, with the union line shown in dashed red. (D) Response CV is plotted against mean response amplitude for each off-step duration, for each cone. Temporal precision of responses were measured as time to threshold (E) and temporal jitter around threshold (F), here measured as SD of time to threshold over 25 stimulus repetitions.

3. From the cone responses to 20Hz flickering stimuli, it seems that some cone output synapses can reliably track signals and may exhibit a faster rate of recovery from synaptic

depression. Can this timescale of synaptic recovery be measured by paired light flashes of varying intervals?

Thank you for suggesting these experiments. We have carried them out and they are reported in the new Figure 6 and associated text where we conclude that “This heterogeneity in the dynamics of refilling likely contributes to wide variations in temporal filtering and jitter across cones.”

4. Does the strength of temporal tuning correlate with the DLI across cone output synapses?

Thanks, yes it does!

While we did not record cone responses to both the chirp stimulus and the varying contrast stimulus (to calculate temporal tuning and DLI, respectively), we could nevertheless calculate release at mean light levels (which correlates strongly with DLI, see Fig 6E) from responses to the chirp stimulus. Re-analysis of this data shows that release at mean light levels within the dynamic range of the cone does correlate with temporal tuning (Reviewer Figure 3, below).

Specifically, mean glutamate release during the 2 s of light on, light off, and light at mean light levels presented at the start and end of the chirp stimulus was measured. The mean values for light on and light off provide an approximate dynamic range for the cone, and the position within which baseline release sits within this dynamic range can then be calculated. Plotting baseline release within the normalised dynamic range against spectral tuning reveals a positive correlation between the two metrics, that is, the higher baseline release sits, the higher the spectral tuning of a given cone (Pearson correlation, $r = 0.598$, $p < 0.001$). The results indicate that higher baseline release is linked with an ability to transmit information about higher frequencies.

Reviewer figure 3 | Higher baseline release predicts tuning to higher frequencies. Baseline release was calculated as the mean glutamate release during exposure to mean light levels within the normalised dynamic range of the cone, giving a value between 0 and 1. When plotted as a function of spectral centroid, extracted from

responses to a 20 Hz chirp stimulus, baseline release forms a positive correlation (Pearson correlation, $r = 0.598$, $p < 0.001$) for $n = 41$ red-cones, 12 fish.

5. HC feedback should also affect the kinetics of cone output responses to light decrement and increment flashes. This should be tested in addition to any potential effect of HC feedback on luminance coding, reliability and temporal precision of cone output.

Thanks, yes, they do!

Specifically, we now compare response kinetics of cone glutamate release to positive and negative responses before and after HC block. For simplicity, we show the responses to 100% positive and 100% negative contrast step conditions (see reviewer Figure 4 below).

Time to threshold and temporal jitter around threshold were measured around a threshold. For responses to -100% contrast which trigger an increase in glutamate release, the threshold was mean glutamate release at mean light levels + 1 sd, and for responses to +100% contrast which trigger a decrease in glutamate release, the threshold was mean - 1 sd. The lower threshold used here (compared to the usual mean + 2 SD) facilitated better capture of the typically smaller, hyperpolarising responses to +100% contrast in CNQX.

We find that after HC block, time to threshold significantly increases for both increments and decrements of light; for -100% contrast shifting from 20.8 to 25.6 ms ($p < 0.001$) and for +100% contrast shifting from 42.2 to 50.3 ($p < 0.05$). Under these conditions, blocking HC feedback causes a delay in responses to light increments and decrements. HC feedback accelerates glutamate release, tightly coupling the glutamate release event with the onset of the contrast stimulus. The temporal precision of responses to -100% contrast was not significantly altered after HC block, but temporal jitter does significantly increase after HC block for +100% contrast responses ($p < 0.05$). Note, the small, decrements in glutamate release in the +100% CNQX condition are typically dominated by noise, and therefore a single point readout such as time to threshold is likely unreliable. In other contrast response analyses, we have used AUC or the average over a time window to circumnavigate this issue which particularly affects On-contrast steps under the CNQX condition.

Amplitude was measured as the mean response during the transient window. We find that absolute amplitude significantly increases for -100% contrast responses ($p < 0.001$), but not for +100% contrast responses. Amplitude reliability, here measured as the coefficient of variance, does not significantly change for either condition.

Together the data demonstrate that HC feedback significantly accelerates responses to increments and decrements of light and tempers the amplitude of responses to -100% contrast. Under these conditions, HC feedback appears to have no significant influence on amplitude reliability, nor temporal precision of -100% contrast responses.

The changes in luminance coding are already shown in Fig 6, where the DLI is shown to decrease after HC block, reflecting a stronger bias towards encoding negative contrasts in the absence of HC feedback.

Reviewer figure 4 | HC feedback accelerates glutamate release. **A**, Mean glutamate release in response to 10 repetitions of the -100% contrast step is shown for an example cone. Note, after HC block, the response is delayed (red). Histograms showing a significant increase in time to threshold (mean + or - 1 SD) for individual cones before (black) and after (red) HC block for both -100% contrast (**B**) and +100% contrast (**C**) steps. **C**, No significant difference is observed in temporal jitter of responses to -100% contrast, but is for responses to +100% contrast, **E**. **F**, Peak response amplitudes are marked (blue dots) for the example cone's responses to -100% (top, left), +100% (bottom, left) contrast steps in control conditions, and to -100% (top, right) and +100% (bottom, right) contrast steps after CNQX injection. **G**, Mean transient amplitude to -100% contrast significantly increases after HC block, but changes in the coefficient of variance are not significant, **H**. Transient amplitude, **I**, and response reliability, **J**, for responses to +100% contrast do not significantly change after HC block. Statistics: non-parametric Wilcoxon Signed-Rank Sum Test, significance defined as $P < 0.05$. $N = 27$ red-cones, 17 fish.

6. The DLI of cone output should be highly dependent on the background light level. More hyperpolarized cones with lower baseline release rate should have a more negative DLI. This should be directly shown. This also suggests that the effect of HC feedback on DLI will be less at lower or moderate background light level compared to brighter backgrounds.

Thank you. We now performed new experiments that directly confirm this notion.

Specifically, similar to new Supplemental Figure S4, PR1 glutamate responses to three repeats of pseudo-randomised contrast steps of light were measured at three different background light intensities. The recordings were continuous, and an electronic filter wheel containing neutral density filters positioned in the stimulus light path generated different background light intensities for each repeat of the contrast stimulus (A). The ND 1.0 (10-fold decrease), ND 0.5 (3.2-fold decrease) and ND 0 (no filter, maximum LED brightness) evoked different mean glutamate release rates, visibly decreasing the higher the background light level (A - D).

Under these conditions, red-cones did not reliably produce responses clear enough to accurately calculate dark-light index (DLI) for each light condition for each cone. Therefore, it was not possible to measure how DLI changes as a function of background light level. In place of DLI, we used baseline release within the cone's response range. Baseline strongly and positively correlates with DLI (see Fig. 6E), so can serve as a proxy for DLI.

At the dimmest background light condition, baseline release sits high within the response range of the cone, compared to baseline sitting lower within the response range at the two brighter background light level conditions (C,D, and E). This predicts that at dimmer light levels, cones will have a more balanced response to dark and light contrast steps. At brighter light levels, where baseline is low, cones are likely to have a stronger bias towards dark contrasts, with a compressed dynamic range for encoding positive contrast steps. Functionally, the reduced glutamate baseline reflects the loss of positive contrast responses, and a compression of the portion of the dynamic range used for responding to positive contrasts.

Overall, we have shown that the contrast response functions of cones depend on background light level, and indeed at brighter light levels in a more hyperpolarised state, cones exhibit a lower absolute baseline release rate, and a lower baseline release rate within the response range predicting a low DLI and bias towards negative contrasts.

We agree that under lower light conditions, the effect of HC feedback on DLI will likely be less than observed at bright light levels. Under bright light conditions, negative feedback from HCs should be strongly activated, pushing back against the hyperpolarising effect of the bright light. Under dimmer light conditions, HC negative feedback should be less, as the hyperpolarising effect of the light is weaker. Therefore, whilst we see a significant drop in DLI (a stronger dark bias) when comparing contrast responses before and after pharmacological HC block (see Fig. 6), it is likely that at dimmer light levels where HC feedback is likely weaker, the predicted drop in DLI would be smaller.

Overall, the conclusions from these new experiments are strongly in line w the those shown in new Supplemental Figure S7, For reasons of brevity we have therefore opted exclude these additional results into the present MS.

Reviewer figure 5 | PR1 exhibit a stronger dark bias exposed to bright light. **A**, Positive and negative contrast steps from a mean light level (500 ms steps with 500 ms ISI, 4 contrast conditions; +/- 50 and 100 %, each condition repeated 5 times) were delivered at 3 background light levels; neutral density filters (ND) produced a ~10-fold decrease (ND 1.0, labelled in black) and a ~3.2-fold decrease (ND 0.5, labelled in red) from full LED brightness (no ND filter, ND 0, labelled in blue). The deconvolved glutamate response of an example red-cone is shown. The mean responses to each stimulus condition are shown for the example cone under the ND 1.0 condition (**B**), ND 0.5 condition (**C**) and ND 0 condition (**D**). For each background light condition, the absolute baseline release (calculated as mean glutamate release during 4 s of continuous light exposure prior to contrast steps) is marked by a dashed line, and the response range of the cone under each light condition is marked by the vertical line (calculated as the difference between the mean minimum and mean maximum response during contrast steps). **E**, Absolute baseline release significantly decreases as mean light levels increase, as shown for $n = 144$ PR1, 7 fish (pairwise comparisons of baseline release is significant for ND 1.0 and ND 0.5, ND 1.0 and ND 0, and ND 0.5 and ND 0 (all $p < .00001$ post-hoc Turkey test following a significant one-way ANOVA of $p < .00001$). **F**, Baseline release within the normalised response range for each condition is significantly higher in the dimmest background light condition (ND 1.0 to ND 0.5 and ND 1.0 to ND 0, both $p < 0.0001$), predicting a more balanced dark-light bias. Baseline release within the response range sits lower at ND 0.5 and ND 0, indicating a dark bias in responses under these conditions.